# OpenReview forum: "Investigating Implicit Bias in Large Language Models: A Large-Scale Study of Over 50 LLMs"
_NeurIPS.cc/2024/Workshop/SafeGenAi — SafeGenAi Poster_

### Official Review · Reviewer_Grmk · 2024-10-08
**A review on Investigating Implicit Bias in Large Language Models: A Large-Scale Study of Over 50 LLMs**

**Rating:** 5
**Confidence:** 4

**Review:**

Pros: The study provides a comprehensive evaluation of bias in large LLMs across multiple social domains, such as race, gender, health, and religion. This thorough coverage not only enhances the depth of the bias analysis but also ensures that the findings are applicable in diverse contexts, making an important contribution to understanding LLM fairness.

Cons: While the study discusses societal biases, it could be further enhanced by considering biases inherent in the models themselves, such as position bias, where the order of presented responses affects judgment. Including such intrinsic biases would provide a more comprehensive understanding of implicit biases in LLMs.

---

### Official Review · Reviewer_1j5n · 2024-10-08
**A review of Investigating Implicit Bias in Large Language Models: A Large-Scale Study of Over 50 LLMs**

**Rating:** 5
**Confidence:** 5

**Review:**

Summary: This article mainly discusses implicit bias in LLMs and conducts a large-scale study of more than 50 LLMs. The article points out that even models that perform well in explicit bias evaluation may have implicit bias, and the bias may increase as the size and complexity of the model increases.

**Strengths**:

1. The research extensively covers over 50 language models, analyzing the bias performance across models of different sizes and ages, presenting a vast amount of data.
2. The introduction of Implicit Association Test (IAT) and decision bias assessment methods offers a novel perspective.

**Weaknesses**:

1. The statement that `newer or larger language models do not automatically exhibit reduced bias` (line 258) is a conclusion limited to the specific biases studied in this paper. When selecting prompts for LLMs, **other biases** such as position bias and verbosity bias should also be considered. Drawing conclusions solely from this paper is somewhat premature.
2. Although the paper presents the formula for the self-attention mechanism, it does not establish a correlation with bias examination. I question the **necessity** of including this formula in the text.
3. The variance in the distribution of experimental results is excessively large, yet the number of datasets is **not specified**. I reasonably suspect that the small number of data entries may have caused the significant result deviation.
4. The article does not detail the **experimental setting parameters**, such as temperature. When facing bias interference, using a higher temperature can result in different judgment outcomes even with the same prompt input. The article should clearly state how this issue is addressed.

---

### Official Review · Reviewer_GvX3 · 2024-10-08
**Important paper that lacks a clear message**

**Rating:** 4
**Confidence:** 4

**Review:**

This paper explores the relevant topic of implicit bias accross a wide range of LLMs. The authors speculate that synthetic data in the training phase may play a major role.

However, the paper in its current form precludes acceptance. The main claim is indeed very interesting but weakly supported across the paper. The paper is well structured and performs a systematic analysis across diverse LLMs. This is clearly a strength, and I encourage the authors to continue investigating in order to provide compelling evidence that evaluates the relationships between synthetic data and bias.

Providing such evidence can shape the paper into an impactful contribution to the field.